# *Coriandrum sativum* L.—Effect of Multiple Drying Techniques on Volatile and Sensory Profile

**DOI:** 10.3390/foods10020403

**Published:** 2021-02-12

**Authors:** Jacek Łyczko, Klaudia Masztalerz, Leontina Lipan, Hubert Iwiński, Krzysztof Lech, Ángel A. Carbonell-Barrachina, Antoni Szumny

**Affiliations:** 1Department of Chemistry, Wrocław University of Environmental and Life Sciences, 50-375 Wrocław, Poland; hubert.iwinski@upwr.edu.pl (H.I.); antoni.szumny@upwr.edu.pl (A.S.); 2Institute of Agricultural Engineering, Wrocław University of Environmental and Life Sciences, 51-630 Wrocław, Poland; klaudia.masztalerz@upwr.edu.pl (K.M.); krzysztof.lech@upwr.edu.pl (K.L.); 3Department Agro-Food Technology, Escuela Politécnica Superior de Orihuela, Universidad Miguel Hernández de Elche, Elche, 03312 Alicante, Spain; leontina.lipan@goumh.umh.es (L.L.); angel.carbonell@umh.es (Á.A.C.-B.)

**Keywords:** coriander, HS-SPME-GC/MS, napping, VOCs, chemistry behind quality, OACs

## Abstract

*Coriandrum sativum* L. is a medicinal and aromatic plant spread around the world, with beneficial properties that are well recognized. Both coriander seeds and leaves are used for pharmaceutical and flavoring purposes. Even though coriander seeds tend to be more popular, the leaves are receiving a consistently growing interest, especially because of popularization of Mexican, Northern African, and Indian cuisines. This increased attention brings about the necessity for providing the product with guaranteed quality, which will retain its valuable characteristics, even after post-harvest treatment. For this reason, it is highly necessary to determine reliable protocols for cilantro preservation, which usually include drying procedures; in order to identify the optimal drying treatments, a spectrum of drying techniques—convective, vacuum-microwave, and a combination of convective and vacuum-microwave—were evaluated. Cilantro-based dried products were examined from the perspectives of volatile organic constituent composition and sensory quality. After headspace solid-phase microextraction-GC/MS analysis and sensory tests, the results demonstrate that convective drying at 70 °C for 120 min followed by vacuum-microwave drying at 360 W and convective drying at 70 °C were the optimal drying methods for preserving cilantro aroma quality, while convective drying at 70 °C for 120 min followed by convective finishing drying at 50 °C decreased cilantro aroma quality.

## 1. Introduction

Coriander (*Coriandrum sativum* L.) is a well-known medicinal and aromatic plant (**MAP**) from the Apiaceae family, widely cultivated in North and South America (Canada, Mexico, Guatemala, and Argentina), Central and Eastern Europe (Poland, Czechia, Slovakia, Hungary, and Romania) and Asia (Turkey, Iran, Pakistan, and India) [1,2]. The plant is characterized by globular-shaped seeds (fruits) abundant in essential oils (EOs) in which linalool (57.5–75.1%), geranyl acetate (8.9–24.5%), α-pinene (2.3–23.2%), terpineol (0.08–5.3%), geraniol (0.5–2.3%) and citronellol (0.6–1.6%) are indicated as major constituents [1,3]. The total EO yield of coriander seeds ranges between 0.8 and 2.1% [4]. Another extremely valuable part of coriander is the leaves, also called cilantro, the EO content of which varies between 0.1 and 0.29%, and the major constituents of which are (*E*)-dec-2-enal, (*E*)-dodec-2-enal, decanol, dodecanol, *n*-tetradecanol and decanal [5,6]. Both coriander seeds and cilantro, are used as spices, especially in Central and South America, the Middle East and Asia [7], regarding their flavoring and health beneficial properties. The complex chemical composition of coriander plant provides extensive bioactive activities, such as antioxidant, anti-inflammatory, antimicrobial, and analgesic activities [8,9,10].

As a spice, especially in Europe, cilantro seems to be less popular than coriander seeds; however, the cuisines of Mexico, Tex-Mex, Northern Africa, and India value cilantro’s distinct aroma and utilize it as a fresh, chopped spice or as a dried, crushed ingredient of spice blends (masala, curry blends, curcuma blends, and more) [7]. The overall aroma of two commonly used cilantro varieties—*C. sativum* L. var. *vulgare alef* and *C. sativum* L. var. *microcarpum DC*—is characterized by floral, spicy, pleasant, grassy, herbal, and earthy notes [11]. Cilantro’s odor-active compounds (**OACs**) mainly include (*Z*)-3-hexenal (green, cut-grass notes), decanal (green, citrus-peel notes), (*E*)-2-decenal (green, cut -grass, lettuce notes), (*E*)-2-dodecenal (green, waxy notes) and other aliphatic hydrocarbons [12]. Those subtle aroma features [13], along with widely reported health promoting properties [14], make cilantro a highly desired MAP.

Undoubtedly, fresh, unprocessed material is the most valuable source of the bioactive and aromatic compounds characteristic of cilantro. Nevertheless, in light of the complicated production and transportation chains, it is impossible to always use it as a fresh ingredient; therefore, it is extremely important to assure the guaranteed quality and safety of cilantro-based products. The most essential quality determinants for MAPs are cultivation conditions (including agronomic practices and weather conditions), harvest time, post-harvest treatments (including material preservation), and plant chemotype [15,16]. Among indicated determinants, post-harvest treatment in the form of drying is the most common processing used for preservation of MAPs quality in a long term [17,18]. The latest studies focused on developing cost-effective, thermo-based drying procedures [19,20], which need to be evaluated on various plant materials [21,22,23,24,25,26,27,28]. The greatest interest is focused on convective drying (**CD**), microwave drying, vacuum-microwave drying (**VMD**), freeze-drying, infrared drying, or combined methods such as convective pre-drying followed by vacuum-microwave finish-drying (**CPD**-**VMFD**).

In addition, coriander seeds and cilantro were the objects of numerous studies which considered drying proceedings and their influence on the quality of obtained dried products. Some of them were mainly focused on mathematical modelling within cost and energy efficiency evaluation of the drying process [29,30,31], while other studies included evaluation of drying influence on quality in light of volatile organic constituents (**VOCs**) and non-volatile constituent composition, or even the sensory properties of dried products [32,33,34]. Unfortunately, none of these studies considered a comprehensive experiment design, the intention of which would be to show a complete picture of various drying methods’ influence on VOC composition and their impact on aroma sensory quality of coriander leaves. The most extensive study was designed by Pirbalouti et al., (2017) [32], since it included five drying methods and their influence on cilantro EO composition; however, sensory evaluation was not performed. On the other hand, Fathima et al. (2001) [34] provided a detailed sensory evaluation of cilantro, without testing a wide spectrum of drying methods and VOCs analysis. Therefore, the aim of this study was to investigate the influence of various drying methods on the sensory quality of cilantro and its VOC composition. For this purpose, CD, VMD, and CPD-VMFD with various parameters were applied to obtain dried cilantro-based products, which were evaluated with headspace solid-phase microextraction-GC/MS (HS-SPME-GC/MS) analysis and multiple sensory analyses.

## 2. Materials and Methods

### 2.1. Plant Material

Cilantro (*Coriandrum sativum* L.) plants were delivered by Swedeponic Polska Sp. z o.o. company (Grodzisk Mazowiecki, Poland). The used plants were of a flat-leaf, parsley-like variety, purchased as potted plants (green parts weight 30 ± 2 g). Coriander leaves, immediately after delivery, were chopped, thoroughly mixed, and split for specific purposes—(i) drying procedures, (ii) chemical analyses of fresh material, and (iii) sensory analyses of fresh material. Plant material was vacuum packed and stored at −18 °C awaiting further proceedings.

### 2.2. Chemicals and Reagents

The analytical grade cyclohexane used during analyses was obtained from UQF (Wrocław, Poland). 2-undecanone, C_7_-C_40_
*n*-alkanes mixture and pure analytical standards (such as decanal, β-pinene, (*Z*-hex-3-en-1-ol, linalool, (*E*)-dec-2-enal, (*E*)-dodec-2-enal, (*E*)-tridec-2-enal, (*E*)-tetradec-2-enal and (*Z*)-tetradec-2-enal) were purchased from Sigma-Aldrich (Saint Louis, MO, USA).

### 2.3. Drying Methods

Two drying methods (convective drying, vacuum-microwave drying) and their combinations were utilized [35]. In each case, 80 g of fresh cilantro samples were used. Moisture content (**Mc**) of the samples was determined using a vacuum dryer pre-set at 70 °C and 100 Pa (SPT-200, ZEAMIL Horyzont, Krakow, Poland). Initial moisture content of the material was Mc = 92.62%. The maximum temperature of the sample was determined after each measurement using an infrared camera (Flir Systems AB, Stockholm, Sweden), right after the removal from the dryer. All drying variants were carried out in two technological repetitions and the process lasted until the moisture content was below 10% wet basis (wb).

#### 2.3.1. Convective Drying (CD)

Convective drying was performed at 50, 60 and 70 °C using the equipment engineered at the Institute of Agricultural Engineering (Wroclaw University of Environmental and Life Sciences, Wroclaw, Poland). In addition, two-stage convective drying with temperature adjustment after 120 min was performed (Table 1). The air velocity in each case was 0.5 m/s.

#### 2.3.2. Vacuum-Microwave Drying (VMD)

Vacuum-microwave drying was carried out at 240, 360, and 480 W (Table 1). The SM-200 dryer (Plazmatronika, Wroclaw, Poland) was used in the study. The pressure in the dryer ranged from 4 to 6 kPa. The samples were weighed after 4, 3 or 2 min in the case of drying at 240, 360, and 480 W, respectively.

#### 2.3.3. Combined Convective Pre-Drying Followed by Vacuum-Microwave Finishing Drying (CPD-VMFD)

Combined drying consisted of 120 min of convective pre-drying at 50, 60 or 70 °C followed by vacuum-microwave finishing drying at 360 W (Table 1) carried out until desired moisture content was obtained.

### 2.4. Modelling of Drying Kinetics

The drying kinetics were fitted on the basis of mass losses recorded during drying by different methods. Moisture ratio (**MR**) was calculated according to the simplified equation:(1)$$MR=\frac{M_{\left(t\right)}}{M_{0}}$$
where *M_(t)_* is the moisture content at a given time and *M_0_* is the initial moisture content.

The drying model was fitted using Table Curve 2D software (Systat Software, San Jose, California, USA) according to the experimental points obtained in the study. The good fit of the applied model was determined on the basis of the highest values of R^2^ and the lowest values of root mean square error (RMSE). Preliminary studies found that the best fit was obtained when Page’s model was applied:(2)$$MR=A*e^{-k*t^{n}}$$
where *A*, *k*, and *n* are drying constants and τ is the time taken for drying.

### 2.5. VOC Profiling

The VOC profile of cilantro was investigated according to Łyczko et al.’s (2019b) [35] protocol, with later slight modifications made by Łyczko et al. (2020) [36]. Briefly, plant material samples were weighed and placed into headspace vials together with internal standard (**IS**) (2-undecanone), and afterwards VOC extraction was performed with divinylbenzene/carboxen/polydimethylsiloxane (DVB/CAR/PDMS) SPME fiber, needle size 23 ga, StableFlex, for use with manual holder/autosampler, fiber L 2 cm (Supeclo, Bellefonte, PA, USA). Further, GC-MS analysis was carried out on Varian CP-3800/Saturn 2000 apparatus (Varian, Walnut Creek, CA, USA) and analytes were separated by Zebron ZB-5MSi (30 m × 0.25 mm × 0.25 µm) column. GC temperature program: initially 50 °C, then to 130 °C at 4.0 °C/min ratio, then to 180 °C at 20 °C/min ratio; carrier gas: helium with linear velocity 35.0 cm/sec; split ratio 1:10. MS operational conditions: ion source temperature 250 °C; electron impact (EI) ionization at 70 eV; scanning range from 35 to 300 *m*/*z*. Analyses were run in three repetitions.

For qualitative analysis, Varian Workstation software was used, whereas for quantitative analysis, ACD/Spectrus Processor (Advanced Chemistry Development, Inc., Toronto, ON, Canada) was utilized. VOCs were identified via determination of linear retention indices (LRIs) and mass spectra comparison (Adams, 2012 [37]; NIST 17 Mass Spectral and Retention Index Library). The LRIs filter was narrowed down to ±15 points and only mass spectra matches with similarity score ≥90% were qualified as hits. The molecular mass of VOCs was confirmed by chemical ionization mode performed with isobutane as the reagent gas. Quantification was done by peak area normalization against IS peak area.

### 2.6. Sensory Evaluation

The sensory quality evaluation of dried cilantro was carried out, as previously reported by Łyczko et al. (2020) [36], with a trained panel of 20 panellists (age 23–52) from the Food Quality and Safety Group (*Escuela Politécnica Superior de Orihuela*) of the *Universidad Miguel Hernández de Elche* (Orihuela, Alicante, Spain). The selection of panelists was performed according to ISO standard 8586-1 [38,39]. Before the actual evaluation, panelists were involved in 3 training meetings in order to adjust the panelist’s senses for required aroma attributes. Sensory analyses were performed in a tasting room with individual booths, controlled light (70–90 fc), and controlled temperature (22 ± 1 °C). Coded cilantro samples were provided for sensory analysis, during which 3 tests were performed— a ranking test, napping test, and descriptive sensory analysis. In the first place, ranking and napping tests were carried out. The ranking test was used to evaluate the intensity of dried products aroma in comparison to fresh cilantro aroma and to reduce the sample number for further steps. The reduction in samples for descriptive sensory analysis was required, since it was impossible to fully study all samples initially available in this study [40,41]. For this, 20 panelists were asked to rank the coriander samples according to perceived intensity of the attribute “fresh coriander”. The panelists used the ascending rank order, in which 1 meant the least and 13 meant the most intense sample, respectively. For descriptive sensory analysis, one sample was selected from each drying treatment/type, having high intensity in the ranking test and having no measurable intensity of off-flavors. For the descriptive sensory analysis, a sensory sheet was developed containing 21 sensory attributes previously selected from the according to the scientific literature, napping test, and orientation sessions. The samples were served in 30 mL covered cups and a random block design was used for sample presentation in order to avoid biases. Each panelist was asked to smell and score the samples using an 11-point numerical scale, where 0 represented none or no intensity and 10 extremely strong, with a 0.5 increment. Additionally, the napping technique was used to group dried samples and to support the reduction in sample number for descriptive sensory analysis. Napping consisted of the position of the products on a A3 sheet of paper according to their similarities by the panelists, who were also asked to add at least 3 descriptors for each group of samples [42]. As the last step, descriptive sensory analysis was carried out by 7 judges who evaluated 4 chosen samples. In this step, the panelists evaluated the basic tastes, aroma, and flavor of dry cilantro according to the lexicon first given by Łyczko et al. (2020) [36] with slight modifications (the full lexicon is available in the Appendix A).

### 2.7. Statistical Analysis

Statistical analysis was performed with Statistica 13.3 software (StatSoft, Kraków, Poland). The one-way analysis of variance (ANOVA) of HS-SPME-GC/MS data and descriptive sensory analysis were carried out, followed by Tukey’s test (*p* < 0.05), and for a ranking test, Friedman’s rank-sum analysis (α = 0.05), supported by Tukey’s honest significance difference (*p* < 0.05), was used. The statistical analysis and graphic interpretation (biplot) of the napping results were carried out with XLSTAT Premium 2016 software (Addinsoft, París, France). The procedure of data handling, collected during napping, consisted of determining the position of each sample on X and Y axis and the sensory description given by each panelist. Check-all-that-apply (CATA) and multifactor data analysis (MFA) analyses were performed to generate the biplot. In all relevant cases, standard deviation (SD) was given.

## 3. Results

### 3.1. Drying

Figure 1 shows the drying kinetics of the cilantro samples treated by different drying methods and Table 2 shows the drying kinetics constants (*A*, *k*, *n*), drying times, maximum temperature of sample heating (*T_max_*), and final moisture content (*Mc_wb_*) of cilantro dried using different methods. It can be seen that during CD with an increase in temperature from 50 to 70 °C, the duration of drying decreased over two times (from 570 min in the case of CD50 to 270 min in case of CD70). On the other hand, application of two-term CD led to an only slightly shorter time of the process than in case of CD50, and even longer times than in case of CD60 or CD70. Furthermore, CD with temperature adjustment after 120 min resulted in higher moisture content than in the case of CD60 or CD70. A smooth transition between temperatures in two-term CD can be noticed, except for CD50/70, where a sudden drop in MR value after 120 min can be seen (Figure 1b).

The highest Mc_wb_ was obtained when CD70/60 was used and the highest water reduction, and therefore the lowest Mc, was reached when CD60 was applied. As for VMD, an increase in the power of magnetrons from 240 to 480 W significantly reduced the duration of the process (from 68 min in the case of VMD 240 to 42 min when VMD 480 was used). Overall, VMD resulted in much shorter processing time comparing to convective drying. Furthermore, the maximum temperature of the sample treated by VMD increased when higher power was applied.

Combined drying resulted in shorter time of the process than in the case of CD, but still higher than when only VMD was applied. The use of this method resulted in moderate final Mc of the sample as well as moderate maximum temperature of the material recorded during drying. Page’s model was successfully applied in the study to describe drying kinetics on the basis of high values of determination coefficient (*R^2^* > 0.99 in every case) and low values of RMSE (RMSE < 0.026). This section may be divided by subheadings. It should provide a concise and precise description of the experimental results, their interpretation, as well as the experimental conclusions that can be drawn.

### 3.2. Aroma Profiling

In total, 61 VOCs were found in the fresh cilantro VOC profile composition, and 59 of them were successfully identified (the mass spectra of the two unknown compounds are given in the Appendix A). As presented in Table 3, (*Z*)-3-hexen-1-ol (17.38 ± 0.38%), nonane (9.28 ± 1.20%), (*Z*)-3-hexen-1-ol acetate (34.54 ± 2.08%), limonene (2.43 ± 0.22%), linalool (1.44 ± 0.02%), decanal (4.53 ± 0.13%), decanol (4.76 ± 0.56%), dodecanal (1.76 ± 0.30%), (*E*)-2-dodecenal (4.18 ± 0.36%), dodecanol (1.35 ± 0.67%) and (*Z*)-2-tetradecenal (4.77 ± 0.50%) were reported as the most abundant VOC profile constituents. The rest of the VOCs present in cilantro VOC profile occurred in quantities lower than 1%.

### 3.3. Influence of Drying on Cilantro Volatile Organic Constituents Composition

Table 4 illustrates that applying various drying methods caused a significant shift in all OACs’ contributions in the cilantro VOC profile. Regarding the most abundant cilantro OACs (more than 1% of contribution)—(*Z*)-hex-3-en-1-ol (17.38%), linalool (1.44%), decanal (4.53%, decanol (4.76%), dodecanal (1.76%), (*E*)-dodec-2-enal (4.18%) and (*Z*)-tetradec-2-enal (4.77%)—only the share of (*Z*)-hex-3-en-1-ol decreased in all dried products below 1%. For other most abundant OACs, the pattern of changes was not as clear—particular OACs’ contribution shifts were strongly dependent on applied drying method. The most interesting observation was that, except for (*Z*)-hex-3-en-1-ol, the contribution of all cilantro OACs was significantly increased by applying drying, regardless of the used method. In addition, it is worth underlining is that drying methods including only VMD caused a significant increase in β-pinene and terpinolene contribution, which are responsible for moldy, earthy, and mushroom aroma notes of dried products.

### 3.4. Sensory Analysis

Multiple sensory analyses showed a significant divergence, regarding dried cilantro odor quality on various levels. The very first trial—the ranking test, the results of which are presented in Table 5—evaluated the intensity of the cilantro-like aroma of dried samples. Samples CPD70-VMFD (rank sum 198 points) and CD70 (sum 184 points) were recognized as those with the most intense cilantro-like aroma, while CD50/70 (sum: 100 points), VMD 480 (sum: 98 points) and CD70/50 (sum: 95 points) were pointed out as those with the weakest aroma intensity.

The napping test, the results of which are illustrated in Figure 2, showed that dried cilantro samples may be divided into five groups. Samples were grouped in the napping test according to the groups established in a clustering dendrogram prepared using Pearson’s correlation based on the unweighted average. The two largest groups, A and C, consisted of three and four samples, respectively, which all included the CD method. Group C, with higher CD initial temperatures, was characterized as “dry grass”, “hay-like”, “parsley-like”, “sweet”, and “chamomile”, while group A, where initial CD temperatures were lower, was characterized as “green”, “lacteal”, “fermented grass”, “wet”, “fresh”, “balsamic”, and with “high intensity”. The next groups, D and E, both contained two samples. Group D, consisting of CPD50-VMFD and CD60/70, was characterized only as “dry leaves”, while group E, consisting of VMD 240 and VMD 480, was characterized as “fermented” and “musty”. One sample—VMD 360—was categorized as group B, characterized as “dry”, “woody”, “infusion”, and “mushrooms”, while the last sample, CD70/50, overall was not characterized with any descriptions.

As the last sensory test, descriptive sensory analysis was performed. For this trial, due to the high number of samples, only one representative for each drying method was selected after ranking and napping techniques—CD60, CD50/70, VMD 240, and CPD60-VMFD. In the case of basic taste descriptors, samples did not show any significant differences, while in case of flavor, pungency and spiciness were the differentiating features. More significant differences were observed among aroma descriptors. As visualized in Figure 3, chamomile, hay-like, herbaceous, vegetable, and woody were the descriptors, which diversified dried cilantro in the case of aroma. CD60 treatment was the one with the strongest intensity of the above-mentioned attributes.

## 4. Discussion

### 4.1. Drying Kinetics

The effect of the hot air temperature during CD as well as power of magnetrons during VMD was evaluated in the study. It can be seen that with an increase in temperature during CD, the time taken for drying decreases, maximum recorded temperature increases, and the drying rate of the process accelerates, which is consistent with other studies [26,29,44].

A similar effect was seen in case of VMD, where with an increase in the power of magnetrons, the duration of the process decreases. The same phenomena were also noticed in thyme [45]. Furthermore, increased wattage resulted in an increase in the maximum temperature recorded during VMD, which is consistent with other studies [46].

During CD, the hot air flows around the sample and efficiently removes surface moisture, which is directly correlated with the high drying rate at the initial stage of the CD. Yet, as can be seen in the Figure 1a, over time, the drying rate diminished as a result of slow diffusion of the moisture from the inside of the sample. On the other hand, faster evaporation during VMD is due to the heating in the whole volume of the sample as a result of microwave application, which mitigates the limitations of CD.

The beneficial effect of combined CPD-VMD was due to the phenomena occurring during both CD and VMD, where at the initial stage the surface water is efficiently removed by CD and, then, the VMD is applied to remove the remaining water which is strongly bounded to the plant cellular structure [47]. As a result, a shorter drying time can be obtained.

The study on leaf drying revealed that the quality of leaves is highly dependent on the duration of the process. However, higher temperatures might result in the degradation of bioactive compounds [29]. Two-term CD might mitigate the negative effect of high temperature on the quality of the sample since application of higher temperature of hot air during the first stage of drying does not increase the temperature of the sample. This is due to the intensive evaporation that cools down the sample and the dried material never reaches the temperature of hot air. After 120 min, when the temperature is reduced to 60 or even 50 °C, evaporation slows down and the temperature of the sample stays relatively low. In addition, in the case of CD50/70, the temperature adjustment changed the shape of the curve, which is marked by a sudden drop in the MR after 120 min, and significantly shortened the drying process.

Page’s model applied in this study was previously successfully used to describe the drying kinetics of *Origanum majorana* [48] and *Cassia alata* [47].

### 4.2. Volatile Organic Constituents of Cilantro and Quality of Dried Products

The majority of the most abundant VOCs identified in fresh cilantro were qualified as aliphatic aldehydes and alcohols, what corresponds with de Melo et al. (2019) [49], Nurzyńska-Wierdak (2013) [50], and Padalia et al.’s (2011) [51] findings. Nevertheless, some exceptions were found, such as high contributions of (Z)-hex-3-en-1-ol, (Z)hex-3-en-1 acetate, and limonene—17.38 ± 0.38%, 34.54 ± 2.08%, and 2.43 ± 0.22%, respectively. In addition, in comparison to previous studies on cilantro, the high percentage of linalool was surprising, since it is much more characteristic for coriander seeds; however, some studies, such as the one performed by Shahwar et al., (2012) [51], supported this result. Such results may be reasoned in various plant chemotypes, which is well-documented in the scientific literature for coriander and other MAPs such as chamomile or sweet basil [52,53,54].

In terms of dried cilantro, some earlier studies were conducted; however, none of them included such a wide spectrum of drying methods, evaluated in the light of their impact on the sensory quality of products, as in the present study. Clearly, as shown in Table 5, the highest scores in the ranking test were granted to CPD70-VMFD and CD70 products. Furthermore, those two samples were qualified in the same group (C—“dry grass”, “herbal”, “sweet”, “hay-like”, “parsley-like” and “chamomile”) during the napping test. Regarding the cilantro OACs, according to Cadwallader et al. (1999) [12], (*Z*)-hex-3-enal and (*E*)-alk-2-enals are essential for the characteristic cilantro aroma. Moreover, such results were also confirmed by Eyres et al. (2005) [43]. Therefore, focusing on indicated VOCs, the CPD70-VMFD and CD70 products showed some similarities in VOC composition. Both samples showed not statistically significant differences regarding (*Z*)-hex-3-enal, its derivative (*Z*)-hex-3-en-1-ol, (*E*)-undec-3enal, and (*E*)-tetradec-3-enal contribution, which are responsible for green, cut grass, fruity, solvent-like, chemical, pungent, spicy, aldehydic, and floral aroma notes. On the other hand, the CPD70-VMFD and CD70 products differed in the case of (*E*)-dec-2-enal, (*E*)-dodec-2-enal, (*E*)-tridec-2-enal, and (*Z*)-tetradec-2-enal. A slight difference in (*E*)-dodec-2-enal contribution (in favor of CD70 product over CPD70-VMFD—13.95% to 11.80%) and higher difference in (*E*)-dec-2-enal (in favor of CPD70-VMFD product over CD70 product—9.89% to 4.17%) may be responsible for the marginally higher score of the CPD70-VMFD sample. Such an implication is supported by Tamura et al.’s (2013) [55] findings, which report that among cilantro OACs, (*E*)-dodec-2-enal, due to the low odor threshold, has the most significant impact on cilantro aroma quality. Admittedly, in this case, the highest scored dried product (CPD70-VMFD) had a lower contribution of (*E*)-dodec-2-enal than the second in the rank (CD70); however, CPD70-VMFD had a higher content of (*E*)-dec-2-enal, which has a strongly similar odor description to that of (*E*)-dodec-2-enal (Eyres et al., 2005) [43]. One of the most surprising experimental findings was that during the descriptive sensory analysis, particular aroma attributes of CD60 scored the highest, while during the ranking test, the CD60 sample was placed lower than the CPD70-VMFD and CD70 samples. Such a result supports the hypothesis that the strongest does not always mean the finest, as was proven in our earlier studies on Thai basil and lavender flowers [23,36]. Furthermore, in the napping test, CD60 was qualified into the group with some off-flavor descriptions (“lacteal”, “wet”, “fermented grass”), which also may be the reason for locating this sample lower in the ranking test than CPD70-VMFD and CD70.

The lowest scores in the ranking test were received by CD50/70, CD70/50, and VMD 480 products. Nevertheless, samples were qualified to groups (A and E) or closely placed next to groups (CD70/50) characterized by MAPs with off-flavor descriptions such as “fermented grass”, “wet”, “lacteal”, “mushrooms”, “fermented” or “musty”. Confronting these results with the cilantro VOC profile composition, it was unambiguous that for ”fermented”, “musty”, and “mushroom” descriptions of drying treatments consisting only of VMD, high contributions of β-pinene and terpinolene were responsible, which, according to Breheret et al. (1997) [56], have a significant role in a wild mushroom aroma. In addition, it is interesting to mention that the compound responsible for the woody notes (Figure 3) of dried cilantro was identified as (*Z*)-tetradec-2-enal. There was a clear relationship between the woody notes’ intensity and the contents of this compound in the VOC profile analysis.

In earlier studies by Fathima et al., (2001) [43] and Pirbalouti et al., (2017) [32], it was pointed out that the application of microwave and high temperature (up to 60 °C) during cilantro drying decreased the quality of the dried products, which seems to support the part of obtained results in the present study, in terms of the results obtained in the case of using VMD individually. Overall, the lowest scores were granted to drying methods consisting only of VMD. In addition, VMD received the most negative odor descriptions (mushroom) during the napping test. On the other hand, the previously cited studies pointed out that applying high temperatures may also be destructive for cilantro aroma quality. Nevertheless, the current results show that utilizing only CD gave satisfying results, which corresponds to Kamel et al.’s (2013) [33] findings. The differences in quality of dried cilantro by CD may be caused by many factors, such as the humidity of the environment in which the process was performed, drying air velocity, and more. The overall satisfying results of combined drying, consisting of CPD followed by VMFD, applied for herbs was earlier confirmed in Chua et al., (2019b, 2019a) [47,57], Calín-Sánchez et al. (2013, 2011) [45,58] and Ali et al.’s (2020) [59] studies. Furthermore, Pirbalouti et al. (2017) [32], in their study, based the evaluation of the quality of dried cilantro on EO composition, which, as was proven in our earlier studies on lavender [23,35] and Thai basil [36], is not correct for the determination of the quality for whole plants.

## 5. Conclusions

The growing demand for medicinal and aromatic plants, especially in terms of flavoring purposes, results in an urgent need to determine optimal protocols in the case of preservation. Regarding the cost and time efficiency, drying technologies are leading post-harvest treatments. Earlier studies showed that the universal drying protocol for all kind of medicinal and aromatic plants is questionable, since plants’ morphological differences and specific volatile organic constituent compositions result in different process requirements. In this study, the evaluation of different drying methods for the preservation of *Coriandrum sativum* L. leaves (cilantro) was performed. The experimental results of the cilantro aroma profile (HS-SPME-GC/MS), multiple sensory analyses of dried products (ranking test, napping test, sensory descriptive analysis), and technological aspects of drying processes show that convective drying at 70 °C for 120 min followed by vacuum-microwave drying at 360 W and convective drying at 70 °C were the optimal drying treatments guaranteeing cilantro-based products the highest quality. The aroma profile of these samples was characterized by low contents of (*Z*)-hex-3-enal and (*Z*)-hex-3-en-1-ol and intermediate contents of (*E*)-dec-2-enal, (*E*)-dodec-2-enal, (*E*)-tridec-2-enal, and (*Z*)-tetradec-2-enal. In contrast, the lowest rated samples, mainly including only vacuum-microwave drying and convective drying at two temperatures, namely vacuum-microwave drying with a power of 480 W or convective drying at 50 °C for 120 min followed by convective finishing drying at 70 °C, were characterized by high contents of β-pinene and terpinolene.

## Figures and Tables

**Figure 1 foods-10-00403-f001:**
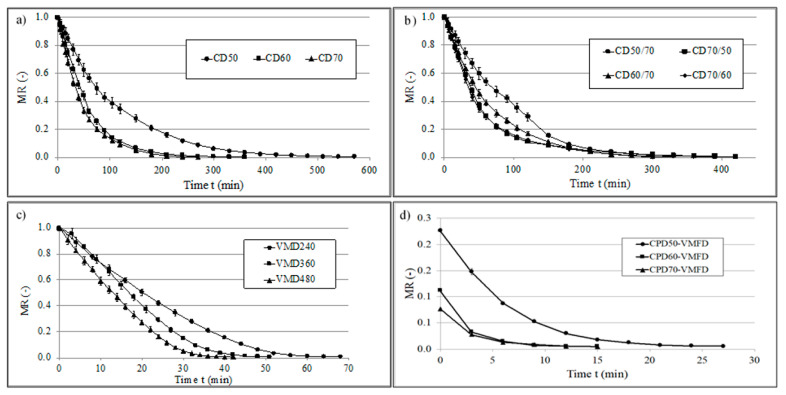
Drying kinetics of cilantro obtained by (**a**) convective drying at 50, 60 and 70 °C; (**b**) convective pre-drying for 120 min at 50, 60 and 70 °C followed by convective finishing drying; (**c**) vacuum-microwave drying at 240, 360 and 480 W; (**d**) combined convective pre-drying at 50, 60 and 70 °C followed by vacuum-microwave finishing drying at 360 W—results presented for VMFD part of the treatment.

**Figure 2 foods-10-00403-f002:**
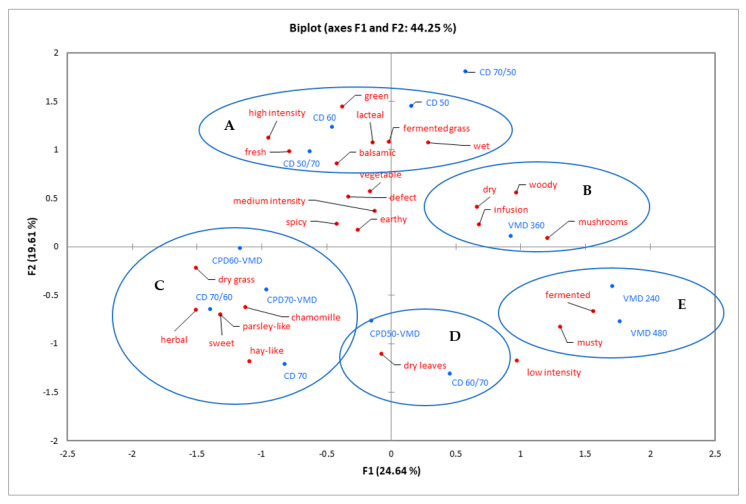
Napping test results for cilantro treated with various drying methods presented as biplot generated with MFA, which explains 44.25% of the variance. Samples were grouped in the napping test according to the groups established in a clustering dendrogram prepared using Pearson’s correlation based on the unweighted average.

**Figure 3 foods-10-00403-f003:**
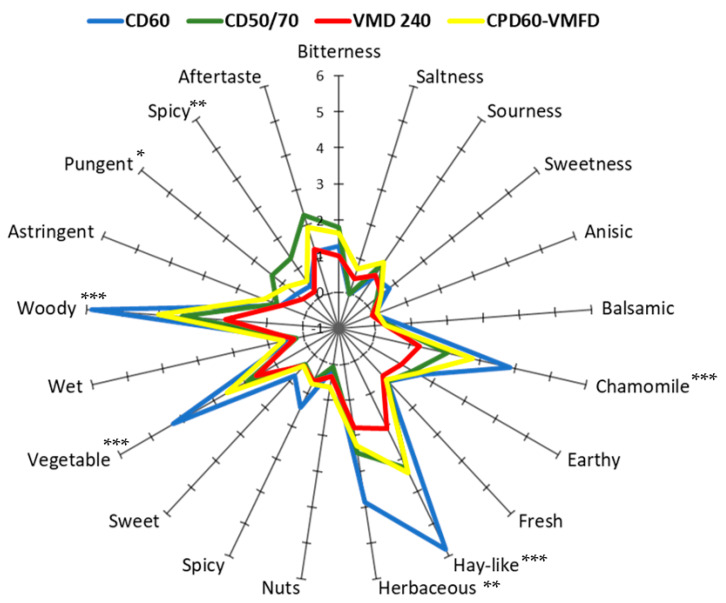
Descriptive sensory results for cilantro samples from each type of drying treatment. The scale used ranged from 0 = no intensity to 10 = extremely strong intensity; *, **, and *** significant at *p* < 0.05, 0.01, and 0.001, respectively.

**Table 1 foods-10-00403-t001:** Dried cilantro samples codes.

Drying Method	Code
Convective drying at 50 °C	CD50
Convective drying at 60 °C	CD60
Convective drying at 70 °C	CD70
Convective drying at 50 °C for 120 min followed by convective finishing drying at 70 °C	CD50/70
Convective drying at 60 °C for 120 min followed by convective finishing drying at 70 °C	CD60/70
Convective drying at 70 °C for 120 min followed by convective finishing drying at 50 °C	CD70/50
Convective drying at 70 °C for 120 min followed by convective finishing drying at 60 °C	CD70/60
Vacuum-microwave drying with power 240 W	VMD 240
Vacuum-microwave drying with power 360 W	VMD 360
Vacuum-microwave drying with power 480 W	VMD 480
Convective pre-drying at 50 °C for 120 min followed by vacuum-microwave drying at 360 W	CPD50-VMFD
Convective drying at 60 °C for 120 min followed by vacuum-microwave drying at 360 W	CPD60-VMFD
Convective drying at 70 °C for 120 min followed by vacuum-microwave drying at 360 W	CPD70-VMFD

**Table 2 foods-10-00403-t002:** Model constants (*A, k, n*), drying times, maximum temperature (T_max_), and final moisture content (Mc_wb_) of the cilantro samples dried using different methods.

Drying Conditions	Constants			Statistics		Drying Time [min]			Maximum Temperature	*Mc*_wb_ [%]
	*A*	*k*	*n*	RMSE	*R^2^*	CPD	CD	VMD	T_max,_ [°C]	
CD50	1	0.0091	0.995	0.0115	0.9989	-	570	-	50 ± 2	6.2 ± 0.5
CD60	1	0.0106	1.124	0.0206	0.9964	-	360	-	60 ± 2	3.3 ± 0.2
CD70	1	0.0192	1.021	0.0109	0.9989	-	270	-	70 ± 2	3.6 ± 0.3
CD50/70	1	0.0062	1.123	0.0188	0.9972	120	270	-	70 ± 2	4.0 ± 0.3
CD70/50	1	0.0151	1.062	0.0215	0.9958	120	300	-	50 ± 2	8.1 ± 0.5
CD60/70	1	0.0151	1.001	0.0075	0.9995	120	180	-	70 ± 2	6.4 ± 0.4
CD70/60	1	0.0193	1.002	0.0227	0.9953	120	210	-	60 ± 2	9.7 ± 0.6
VMD 240	1	0.0111	1.391	0.0242	0.9943	-	-	68	50 ± 2	9.0 ± 0.5
VMD 360	1	0.0063	1.671	0.0143	0.9982	-	-	51	53 ± 2	7.8 ± 0.3
VMD 480	1	0.0191	1.441	0.0264	0.9932	-	-	42	56 ± 2	8.1 ± 0.2
CPD50-VMFD	0.2231	0.1351	1.082	0.002	0.9992	120	-	27	53 ± 2	6.1 ± 1.1
CPD60-VMFD	0.1111	0.5842	0.687	0.0014	0.9982	120	-	15	55 ± 2	5.9 ± 0.9
CPD70-VMFD	0.0768	0.4921	0.697	0.0012	0.9972	120	-	15	62 ± 2	5.2 ± 0.9

CD—convective drying, CPD—convective pre-drying, VMD—vacuum-microwave drying, VMFD—vacuum-microwave finish-drying.

**Table 3 foods-10-00403-t003:** Volatile organic constituent composition in fresh cilantro aroma profile.

Compound	Compound Class	LRI_exp_ ^1^	LRI_lit_ ^2^	LRI_lit_ ^3^	Contribution ^4^ [%]	The Match Fitting Score ^5^ [%]	Odour Description ^6^
3-Methyl-1-butanol	alcohol	736	736	731	0.34 ± 0.01	93	
2-Methyl-1-butanol,	alcohol	739	739	-	0.57 ± 0.01	91	
(*Z*)-Hex-3-enal	aldehyde	803	810	797	0.16 ± 0.04	91	Green/Floral
(*Z*)-Hex-3-en-1-ol	alcohol	851	857	850	17.38 ± 0.38	94	Green, cut grass
(*E*)-Hex-2-en-1-ol	alcohol	863	862	-	0.39 ± 0.02	92	
3-Methylbutyl acetate	ester	874	876	869	0.12 ± 0.01	92	
Allyl Isothiocyanate	other	884	887	-	0.17 ± 0.03	95	
Nonane	alkane/alkene	902	900	900	9.28 ± 1.20	90	
α-Pinene	terpene/terpenoid	934	937	932	0.37 ± 0.05	95	
Camphene	terpene/terpenoid	950	952	946	tr ^7^	94	
2-Methylpropyl butanoate	ester	955	955	-	0.16 ± 0.02	91	
Sabinene	terpene/terpenoid	974	974	969	0.11 ± 0.01	93	
β-Pinene	terpene/terpenoid	979	978	974	0.14 ± 0.01	94	Mouldy, earthy/Mushroom
6-Methyl-hept-5-ene-2-one	others	986	988	-	0.10 ± 0.02	90	
Unknown	-	992			0.31 ± 0.01		
Decane	alkane/alkene	1001	1000	1000	0.27 ± 0.04	94	
(*Z*)-Hex-3-en-1-ol, acetate	ester	1009	1005	1004	34.54 ± 2.08	90	
3-Carene	terpene/terpenoid	1012	1011	1008	1.57 ± 0.06	90	
*p*-Cymene	terpene/terpenoid	1025	1025	1022	0.45 ± 0.01	91	
Limonene	terpene/terpenoid	1030	1030	1024	2.43 ± 0.22	91	
(*Z*)-β -Ocimene	terpene/terpenoid	1038	1038	1032	tr	92	
(*E*)-β-Ocimene	terpene/terpenoid	1049	1049	1044	0.15 ± 0.01	94	
3-Methylbutyl butanoate	ester	1056	1056	1049	0.23 ± 0.01	95	
γ-Terpinene	terpene/terpenoid	1060	1060	1054	0.33 ± 0.03	97	
(*Z)*-Sabinene hydrate	terpene/terpenoid	1072	1064	1069	0.59 ± 0.04	93	
Terpinolene	terpene/terpenoid	1090	1088	1086	0.45 ± 0.04	97	Mushroom, truffle/Mouldy, earthy
Tetrahydrolinalool	terpene/terpenoid	1099	1098	1098	0.38 ± 0.02	93	
Linalool	terpene/terpenoid	1101	1099	1095	1.44 ± 0.02	93	Citrusy/Floral
Limonene epoxide	terpene/terpenoid	1137	1133	1137	0.15 ± 0.01	90	
*p*-Menthone	terpene/terpenoid	1157	1153	1148	0.10 ± 0.01	90	
Menthol	terpene/terpenoid	1174	1175	1167	0.51 ± 0.08	93	
Terpinen-4-ol	terpene/terpenoid	1180	1177	1174	0.13 ± 0.01	90	
Dill ether	terpene/terpenoid	1189	1186	1184	0.10 ± 0.01	90	
*p*-Menth-8-en-2-ol	terpene/terpenoid	1197	1195	1187	0.23 ± 0.02	93	
Estragole	other	1201	1196	1195	0.30 ± 0.02	94	
Decanal	aldehyde	1207	1206	1201	4.53 ± 0.13	96	Floral, citronellol/Fruity
Carvone	terpene/terpenoid	1247	1242	1239	0.65 ± 0.09	92	
Linalyl acetate	terpene/terpenoid	1258	1257	1254	0.27 ± 0.01	92	
(*E*)-De-2-cenal	aldehyde	1264	1263	1260	0.46 ± 0.14	97	Coriander/Aldehydic/Pungent, spicy
(*E*)-Dec-9-en-1-ol	alcohol	1270	1262	1263	0.33 ± 0.10	90	
Decanol	alcohol	1274	1273	1266	4.76 ± 0.56	91	Floral/Spicy
Bornyl acetate	terpene/terpenoid	1290	1285	1283	0.26 ± 0.03	92	
Undecan-2-ol	alcohol	1304	1308	1301	0.18 ± 0.05	97	Medicinal/Pungent, spicy
Undecanal	aldehyde	1310	1307	1305	0.29 ± 0.08	95	Fruity/Floral/Spicy
Methyl dodecanoate	ester	1330	1325	1323	0.10 ± 0.02	91	
Terpinyl acetate	terpene/terpenoid	1357	1350	1346	0.20 ± 0.02	90	
(*E*)-Undec-2-enal	aldehyde	1371	1365	-	0.18 ± 0.08	90	Fruity/Solvent, chemical
(*Z*)-Tetradec-2-ene,	alkane/alkene	1380	1378	-	0.16 ± 0.02	91	
Dec-9-enyl acetate	ester	1398	1399	1399	0.10 ± 0.03	92	
Decyl acetate	ester	1402	1408	1407	0.12 ± 0.03	93	
Dodecanal	aldehyde	1414	1409	1408	1.76 ± 0.30	96	Pungent, spicy/Floral, citronellol
Caryophyllene	sesquiterpene	1420	1419	1417	0.22 ± 0.01	95	
Elemene isomer	sesquiterpene	1444	1444	-	tr	92	
(*Z*)-Dodec-2-enal	aldehyde	1460	1467	-	0.3 ± 0.07	92	
(*E*)-Dodec-2-enal	aldehyde	1472	1471	1468	4.18 ± 0.36	95	Coriander/Floral/Pungent
Dodecanol	alcohol	1477	1473	1469	1.35 ± 0.67	95	
Eremophilene	sesquiterpene	1502	1499	-	0.22 ± 0.03	91	
unknown sesquiterpene	sesquiterpene	1572			0.33 ± 0.21		
(*E*)-Tridec-2-enal	aldehyde	1577	1572	-	0.30 ± 0.04	93	Coriander/Pungent, spicy
(*E*)-Tetradec-2-enal	aldehyde	1667	1673	-	0.26 ± 0.10	94	Pungent, spicy/Aldehydic/Floral
(*Z*)-Tetradec-2-enal	aldehyde	1682	1675	-	4.77 ± 0.50	91	Pungent, spicy/Woody
**Compounds group**				**Total contribution [%]**			
Aldehydes				17.19 ± 0.95			
Alkanes/alkenes				9.71 ± 0.81			
Esters				35.47 ± 1.31			
Alcohols				24.39 ± 1.17			
Others				0.57 ± 0.02			
Terpenes/terpenoids				11.01 ± 0.05			
Sesquiterpenes				0.77 ± 0.18			
SUM				99.11			

^1^ Experimentally obtained linear retention indices (LRI); ^2^ LRI according to NIST17 database; ^3^ LRI according to Adams (2012). ^4^ Values based on HS-SPME analysis; ^5^ the match fitting score of obtained mass spectra to mass spectra available in NIST17 database; ^6^ cilantro odor-active compounds (OACs) and their odor description according to Eyres et al. (2005) [43]; ^7^ tr < 0.10%.

**Table 4 foods-10-00403-t004:** Contribution of particular cilantro OACs in fresh cilantro and samples subjected to drying.

Odour Active Compounds	Fresh	CD50	CD60	CD70	CD60/70	CD50/70	CD70/50	CD70/60	VMD 240	VMD 360	VMD 480	CPD50-VMFD	CPD60-VMFD	CPD70-VMFD
	Contribution [%]													
(*Z*)-Hex-3-enal	0.16f ^1^	0.60bc	0.14f	0.41cde	0.61bc	0.35def	0.47bcd	0.65b	0.56bcd	0.65b	1.70a	0.24ef	0.39cde	0.22ef
(*Z*)-Hex-3-en-1-ol	17.38a	0.58bc	0.12d	0.26cd	0.32bcd	0.24cd	0.28bcd	0.30bcd	0.32bcd	0.39bcd	0.64b	0.25cd	0.23cd	0.08d
(*E*)-Hex-2-en-1-ol	0.39b	0.24cde	0.07f	0.22de	0.28cd	0.15ef	0.24cde	0.32bc	0.32bcd	0.25cd	0.75a	0.11f	0.15ef	0.07f
β-Pinene	0.14de	0.33bc	0.09e	0.26cd	0.35bc	0.22cde	0.34bc	0.41b	0.25cd	0.36bc	0.80a	0.17de	0.24cd	0.14de
Terpinolene	0.45bcd	0.45bc	0.11f	0.32cde	0.45bcd	0.30de	0.42bcd	0.55b	0.35cde	0.37cde	0.70a	0.23ef	0.35cde	0.25ef
Linalool	1.44bc	1.21bcde	0.52f	1.20bcde	1.37bcd	1.14bcde	0.77ef	1.58b	0.60f	0.98cdef	2.19a	0.83ef	0.90ef	0.93def
Decanal	4.53g	15.00d	24.88b	15.18d	6.94ef	20.82c	4.94fg	7.92e	0.32h	15.05d	7.82e	25.25b	23.34b	29.10a
(*E*)-Dec-2-enal	0.46e	1.35e	11.90a	4.17d	0.58e	3.91d	0.31e	0.56e	3.41d	3.25d	1.34e	10.85ab	6.60c	9.89b
Decanol	4.76b	2.71^cd^	5.93b	2.17cd	2.03cd	4.84b	1.39d	3.07c	4.53b	8.95a	1.48d	5.61b	4.75b	5.12b
Undecan-2-ol	0.18bcde	0.25bc	0.08e	0.28b	0.23bcd	0.14cde	0.17bcde	0.21bcde	0.24bcd	0.27bc	0.64a	0.11de	0.15cde	0.10e
Undecanal	0.29d	2.23ab	1.55cd	2.47ab	1.54cd	2.70a	1.20d	1.40cd	0.24d	1.42cd	1.81bcd	1.83bcd	1.85bcd	1.90bc
(*E*)-Undec-2-enal	0.18fg	0.74de	3.14a	1.90c	0.18fg	1.88c	0.14g	0.21efg	0.27efg	0.72def	0.86d	2.59b	1.78c	1.65c
Dodecanal	1.76h	5.34bc	4.36de	5.86ab	3.04fg	6.60a	2.08h	2.54gh	0.33h	3.37fg	3.60ef	4.86cd	3.79ef	4.93cd
(*E*)-Dodec-2-enal	4.18fg	7.31e	19.52a	13.95bc	2.96g	14.51b	1.37h	1.14h	0.32h	2.92g	4.34f	14.05bc	13.07cd	11.80d
(*E*)-Tridec-2-enal	0.30f	0.37ef	1.53a	0.94b	0.22f	0.66bcd	0.19f	0.23f	0.71bc	0.30f	0.63cde	0.80bc	0.64cd	0.42def
(*E*)-Tetradec-2-enal	0.26bc	0.80a	0.19c	0.24bc	0.17c	0.18c	0.14c	0.19c	0.35bc	0.19c	0.64ab	0.25bc	0.19c	0.09c
(*Z*)-Tetradec-2-enal	4.77cd	2.34fg	9.46ab	6.05c	1.96fg	8.21b	1.03gh	0.20h	3.24ef	3.18ef	0.99gh	10.76a	5.54cd	4.38de

^1^ Values followed by the same letters are not statistically different in Tukey’s test and one-way analysis of variance.

**Table 5 foods-10-00403-t005:** Ranking test results—intensity of cilantro-like aroma of dried cilantro samples in comparison to fresh cilantro leaves.

Drying Method	Rank ^1^
CPD70-VMFD	198^a 2^
CD70	184^a^
CD50	181^ab^
CPD60-VMFD	179^ab^
CD60	154^abc^
VMD 360	134^bcd^
CD60/70	124^cd^
CD70/60	122^cd^
CPD50-VMFD	121^cd^
VMD 240	120^cd^
CD50/70	100^d^
VMD 480	98^d^
CD70/50	95^d^

^1^ Maximal rank sum was 260; ^2^ different letters after each dried sample indicate a significant difference at *p* < 0.05 in Friedman’s test by the honest significance difference (HSD) test (α).

## Data Availability

Data is contained within the article or Appendix A.

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
