# Peer review of "Coriandrum sativum* L.—Effect of Multiple Drying Techniques on Volatile and Sensory Profile"

_foods, 2021, doi:10.3390/foods10020403_

Round 1

Reviewer 1 Report

Coriandrum sativum L. – Aroma Profile and Sensory Evaluation in Light of Multiple Drying Techniques

Summary

This paper gives an overview of the impact of convective, vacuum-microwave and combinations of both drying techniques on the volatile contents and sensory attributes of dried cilantro. They concluded that convective drying at 70 °C for 120 min followed by vacuum-microwave drying at 360 W and convective drying at 70 °C were the optimal drying treatments guaranteeing cilantro-based products with the highest quality.

General and specific comments

While earlier papers provide rather fragmented results on this topic, the authors tried to give a comprehensive overview of the impact of both drying techniques on the sensorial attributes of dried cilantro. The authors applied convective drying at different temperature regimes, vacuum-microwave drying at different Wattages and combination of both, which gives a whole picture of the possibilities that both techniques provide.

In order to select an appropriate drying methodology a broad range of tests was performed. In a second step the most promising methods were selected and studied more in detail, which is a justifiable work method.

Although the paper shows a lot of results, unfortunately a more thorough and in dept explanation of the results is lacking. An example. CD60 is situated nearby green, lacteal and fermented grass, while CD70 is situated nearby haylike, sweet, chamomille in Figure 2. This difference is caused by the presence and content of several OACs, and these data are shown in the paper. However, the organic reactions that explain the formation, decomposition and conversion of the AOCs are missing, which is a pity.

From a practical point of view this paper certainly provides useful information regarding the drying of cilantro.

Line number:

15: Type fault: Coriandrum sativum

26: GC/MS is a well-known technique and abbreviation, but HS-SPME isn't. Please show the whole name in the abstract.

56: other

59: Isn’t this description too general? Is fresh cilantro really the most valuable source of bio-active and aromatic compounds that exists?

96: Plant material: which parts?

101: reagents: without capital

103: pure and analytical standards: Can you please specify? This is rather vague.

109: Drying methods: How many repetitions were used for each sample taken at a certain time point?

134: Was the table made according to the journal guide lines?

147: A, K and n are explained, but what is the meaning of “tau”?

155: DVB/CAR/PDMS SPME fiber: please provide the full name, brand and type fibre.

183: Please provide the procedure/setup applied during the ranking test.

192: 4, chosen  samples: Remove comma and space

192: On which basis were these 4 samples chosen? Which criteria were applied to select these samples?

201: How was the biplot generated? Which program software was applied? How were the napping results translated into a biplot?

Fig. 1d: Why do these values not start at 1.0? I assume that the graph only shows the second part, namely the vacuum-microwave drying, but this is not explained in the title.

Table 2: The moisture contents show 3 decimal places, while the st. dev. only mentions 1 decimal place, which is not logic. A moisture content having 4 significant numbers and 3 decimal places is not justified.

250: Remove “(E)”

Table 3: I propose to rank the substances by compound class (aldehydes, esters, …) and additionally by concentration within 1 class. This will give a good overview of the type and contents of the different volatiles found.

Table 3: I suggest to apply the IUPAC names wherever possible. For instance line 1: 3-methyl-1-butanol instead of 1-butanol, 3-methyl-. Other example: Isobutyl butanoate instead of butyric acid, isobutyl ester. In some cases the IUPAC names are quite long. E.g. α-pinene. In such cases the trivial name can be used.

Table 3: Which criterion was used to distinguish between OAC_n and the other compounds? Why 17 OACs?

Table 3: <9->Decenyl, acetate ??

Line 275: Does this table depict relative concentration changes as the title mentions, or relative concentrations? How did the authors determine whether the differences are significant, since no standard deviations are depicted in the table.

Table 4: The odor compounds depicted in this table are shown by their OAC-number, and not by their name. Since in the discussion the compound names are given, it is difficult to read and understand the text. For each compound mentioned in the discussion, the reader has to look up the corresponding name in Table 3, and then go to Table 4 to follow the explanation, which is cumbersome. That’s why I propose to mention the name of the odor active compound instead of the OAC-number.

Figure 2: The biplot only explains 24,64% + 19,61% = 44,25% of the differences, which is not high. Is this sufficiently high to have an accurate image of the distribution of the results?

Figure 2: Which criteria were used to group napping results and methods applied? E.g., which criterion was used to make two VMD groups instead of one? And why was CD70/50 excluded from the group containing CD50?

302: The group numbers are not shown in Figure 2.

313: Which criteria were used to select these drying methods? The text only mentions that the selection was based upon the ranking and napping techniques, but does not provide more details.

385: 17.38±0.38%

421-422: This sentence does not seem to be correct.

425: Why “unfortunately”?

425: “Unfortunately, these samples were not grouped together while the napping test.” Is this sentence complete? It seems a part is missing.

436: I don't see that application of a high temperature yields lower quality in this study. CD70 and CD70/60 give good results. In both case also a high temperature was applied. On the other hand lower temperatures (CD50) result in “fermented grass, green, lacteal, wet” sensory attributes.

Author Response

Jacek Łyczko, PhD candidate

Wrocław University of Environmental and Life Sciences

Department of Chemistry

Norwida 25, 50-375 Wrocław , Poland

jacek.lyczko@upwr.edu.pl

Dear Reviewers,

Thank you for your valuable comments regarding manuscript foods-1074621, entitled Coriandrum sativum L. – Aroma Profile and Sensory Evaluation in Light of Multiple Drying Techniques. We appreciate your detailed review and hope that our corrections will find your acceptance.

  1. Reviewer I:
  2. 15: Type fault: Coriandrum sativum.

The typing has been corrected.

  1. GC/MS is a well-known technique and abbreviation, but HS-SPME isn't. Please show the whole name in the abstract.

The HS-SPME abbreviation was changed for whole technique name. Please see line 26.

  1. 56: other

It has been corrected.

  1. 59: Isn’t this description too general? Is fresh cilantro really the most valuable source of bio-active and aromatic compounds that exists?

Thank you for this valuable point of view. Our intention was not to suggest that cilantro is the most valuable source of bio-active compounds among all plant materials. We meant, that fresh cilantro is much more abundant in aromatic and bio-active compounds than processed in post-harvest treatment one. The sentence was rephrased for: “Undoubtedly fresh, unprocessed material is the most valuable source of bioactive and aromatic compounds characteristic for cilantro”. We hope that it will clarify our intentions.

  1. 96: Plant material: which parts?

During the experiments coriander leaves were subjected to drying procedures. The relevant information was added in Material and Methods section. Please see lines 97-98.

  1. 101: reagents: without capital

It has been corrected.

  1. 103: pure and analytical standards: Can you please specify? This is rather vague.

The list of used during GC-MS analyses analytical standards was given in the text. Please see lines 106-107.

  1. 109: Drying methods: How many repetitions were used for each sample taken at a certain time point?

Thank you for this comment. All drying variants were carried out in two technological repetitions. Information has been included in the Materials and Methods section. Please see lines 117-119.

  1. 134: Was the table made according to the journal guide lines?

The table was intended to be made according to journal guidelines, however some editorial mistakes occurred. It has been corrected and adapted to journal guidelines.

  1. 147: A, K and n are explained, but what is the meaning of “tau”?

Thank you for this point. "Tau" is the time of the process. Relevant information has been included in the text. Please see line 154.

  1. 155: DVB/CAR/PDMS SPME fiber: please provide the full name, brand and type fibre.

The details related to SPME fibre were introduced into Material and Methods section. Please see lines 161-163.

  1. 183: Please provide the procedure/setup applied during the ranking test.

Done as requested. Please see lines 195-198.

  1. 192: 4, chosen samples: Remove comma and space

It has been corrected.

  1. 192: On which basis were these 4 samples chosen? Which criteria were applied to select these samples?

For descriptive sensory analysis was chosen one sample from each drying treatment/type, having high intensity in the ranking test and having no measurable intensity of off-flavours, were chosen. The relevant information was added into the text. Please see lines 198-201.

  1. 201: How was the biplot generated? Which program software was applied? How were the napping results translated into a biplot?

The napping results were analyzed and processed with XLSTAT Premium 2016 software was used. The description was added to the text. Please see lines 222-226.

  1. 1d: Why do these values not start at 1.0? I assume that the graph only shows the second part, namely the vacuum-microwave drying, but this is not explained in the title.

The Reviewer interpretation is precise.  The figure title was completed with relevant clarification. Please see line 262.

  1. Table 2: The moisture contents show 3 decimal places, while the st. dev. only mentions 1 decimal place, which is not logic. A moisture content having 4 significant numbers and 3 decimal places is not justified.

Thank you for this perceptive comment. The decimal places were reduced up to one.

  1. 250: Remove “(E)”

Removed.

  1. Table 3: I propose to rank the substances by compound class (aldehydes, esters, …) and additionally by concentration within 1 class. This will give a good overview of the type and contents of the different volatiles found.

Thank you for this comment. We agree that it will be a valuable input to our work. Nevertheless, we do not agree to group the particular compounds in compounds classes, since we want to remain with growing linear retention indices order. We hope that pointing out the class of each compound and addition of rows with contribution amount of each group will satisfy the Reviewer. In all cases when one compound might be qualified to more than one group (e.g. bornyl acetate - terpene/terpenoid or ester) it was qualified for more characteristic and common for group.

  1. Table 3: I suggest to apply the IUPAC names wherever possible. For instance line 1: 3-methyl-1-butanol instead of 1-butanol, 3-methyl-. Other example: Isobutyl butanoate instead of butyric acid, isobutyl ester. In some cases the IUPAC names are quite long. E.g. α-pinene. In such cases the trivial name can be used.

The IUPAC names, wherever possible, were applied.

  1. Table 3: Which criterion was used to distinguish between OAC_n and the other compounds? Why 17 OACs?

The distinguish between OACs and other compounds was based on Eyres, G.; Dufour, J.-P.; Hallifax, G.; Sotheeswaran, S.; Marriott, P.J. Identification of character-impact odorants in coriander and wild coriander leaves using gas chromatography-olfactometry (GCO) and comprehensive two-dimensional gas chromatography–time-of-flight mass spectrometry (GCxGC–TOFMS). J. Sep. Sci. 2005, 28, 1061–1074, what was pointed out in Table 3 footer (please see lines 286-287).

The Eyres et al. work gives an comprehensive investigation on cilantro OACs, which in our opinion gives a good reason to mark those, particular 17 compounds as the most likely to be the cilantro OACs. We hope that this justification will be satisfying for the Reviewer.

  1. Table 3: <9->Decenyl, acetate ??

The compound name has been corrected for “Dec-9-enyl acetate”.

  1. Line 275: Does this table depict relative concentration changes as the title mentions, or relative concentrations? How did the authors determine whether the differences are significant, since no standard deviations are depicted in the table.

Again, thank you for such perceptive point. The Table 4 presents the relative concentrations of particular OACs. The table’s title has been clarified. Please see lines 304-305.

At this point, we decided to not to give the standard deviations (SD) in Table 4. The table already is extensive and we are afraid that addition of SD will complicate the interpretation. The determination of homogenous groups was obtained by applying STATISTICA program, what was pointed out in Materials and Methods section (lines 218-227). The algorithm of the program, during identification of statistically significant differences uses SD so we think that it is justified to present just particular homogenous groups.

  1. Table 4: The odor compounds depicted in this table are shown by their OAC-number, and not by their name. Since in the discussion the compound names are given, it is difficult to read and understand the text. For each compound mentioned in the discussion, the reader has to look up the corresponding name in Table 3, and then go to Table 4 to follow the explanation, which is cumbersome. That’s why I propose to mention the name of the odor active compound instead of the OAC-number.

We agree with this comment. The OAC-number codes were change for names.

  1. Figure 2: The biplot only explains 24,64% + 19,61% = 44,25% of the differences, which is not high. Is this sufficiently high to have an accurate image of the distribution of the results?

Authors agreed with the reviewer comment, and 44.25 % is not the best percentage to explain the variance. However, as it is close to 50% we consider an important percentage to obtain an accurate image about the results distribution (which helped us to reduce the samples number in order to be able to run descriptive sensory analysis). Additionally, other papers have been published in this regard with similar percentages such as:

Tâm MinhLê, FrançoisHusson, SébastienLê. 2016. Digit-tracking: Interpreting the evolution over time of sensory dimensions of an individual product space issued from Napping and sorted Napping. Food Quality and Preference. https://doi.org/10.1016/j.foodqual.2015.07.002

  1. Lê, T.M. Lê, M. Cadoret. 2015. 9 - Napping and sorted Napping as a sensory profiling technique. Editor(s): Julien Delarue, J. Ben Lawlor, Michel Rogeaux, In Woodhead Publishing Series in Food Science, Technology and Nutrition, Rapid Sensory Profiling Techniques, Woodhead Publishing, Pages 197-213, ISBN 9781782422488. https://doi.org/10.1533/9781782422587.2.197.

Young-Kyung Kim, Laureen Jombart, Dominique Valentin, Kwang-Ok Kim. 2013. A cross-cultural study using Napping®: Do Korean and French consumers perceive various green tea products differently? Food Research International, Volume 53, Issue 1,Pages 534-542, ISSN 0963-9969, https://doi.org/10.1016/j.foodres.2013.05.015.

  1. Figure 2: Which criteria were used to group napping results and methods applied? E.g., which criterion was used to make two VMD groups instead of one? And why was CD70/50 excluded from the group containing CD50?

Samples were grouped in the napping test according to the groups established in a clustering dendrogram prepared using Pearson’s correlation based on the unweighted average. This information has been added to the text. Please see Page 11, Lines 326-328.

  1. 302: The group numbers are not shown in Figure 2.

It has been corrected.

  1. 313: Which criteria were used to select these drying methods? The text only mentions that the selection was based upon the ranking and napping techniques, but does not provide more details.

For descriptive sensory analysis was chosen one sample from each drying treatment/type, having high intensity in the ranking test and having no measurable intensity of off-flavours, were chosen. The relevant information was added into the text. Please see lines 198-201.

  1. 385: 17.38±0.38%

It has been corrected.

  1. 421-422: This sentence does not seem to be correct.

We agree with this comment. The sentence was completed with following clarification: ”[…], what also may be the reason of locating this sample lower in the ranking test than CPD70-VMFD and CD70. Please see lines 451-452.

  1. 425: Why “unfortunately”? and 425: “Unfortunately, these samples were not grouped together while the napping test.” Is this sentence complete? It seems a part is missing.

We agree that this statement was not necessary. It has been removed.

  1. 436: I don't see that application of a high temperature yields lower quality in this study. CD70 and CD70/60 give good results. In both case also a high temperature was applied. On the other hand lower temperatures (CD50) result in “fermented grass, green, lacteal, wet” sensory attributes.

Thank you for this comment. In manuscript we stated: “In earlier studies by Fathima et al., (2001) and Pirbalouti et al., (2017), it was pointed out that the application of microwave and high temperature (up to 60 ºC) during cilantro drying had decreased the quality of the dried products […]” –our intention was to highlight that applying the microwaves more than high temperatures has destructive influence on cilantro quality, what was pointed out in the second part of the sentence: “[…] , what seems to support the part of obtained results in the present study.” Nevertheless, we agree that it may be confounding, thus the explanation, that Authors, as “the part of obtained results” meant the results regarding VMD, which overall, clearly had worse results than other samples. Please see lines 468-469.

Sincerely,

Jacek Łyczko, corresponding author.

Reviewer 2 Report

Coriander (Coriandrum sativum L.) is a well-known culinary and medicinal plant from which seeds and leaves (cilantro) are obtained spices and essential oils. Although its seeds are preferred, fresh leaves are also a valuable source of bioactive and aromatic compounds that are eaten in some Countries fresh or dried as a spice, alone or crushed and mixed with other ingredients. Drying technologies are currently the main post-harvest treatments used to preserve its  characteristics. Therefore, in this work, different drying methods for storing coriander leaves were evaluated.

In my opinion overall the work has good merits and is able to capture the interest of readers. It is clearly and properly written, only few mistakes in the text should be corrected (eg: l. 160: ion source temperature -250 °C, or at 250 °C). The number of references (59) is perhaps too high. Authors are therefore advised to reduce the number of citations, perhaps 30 might be acceptable.

Author Response

Jacek Łyczko, PhD candidate

Wrocław University of Environmental and Life Sciences

Department of Chemistry

Norwida 25, 50-375 Wrocław , Poland

jacek.lyczko@upwr.edu.pl

Dear Reviewer,

Thank you for your valuable comments regarding manuscript foods-1074621, entitled Coriandrum sativum L. – Aroma Profile and Sensory Evaluation in Light of Multiple Drying Techniques. We appreciate your detailed review and hope that our corrections will find your acceptance.

  1. only few mistakes in the text should be corrected (eg: l. 160: ion source temperature -250 °C, or at 250 °C).

This one and other small mistakes were corrected

  1. The number of references (59) is perhaps too high. Authors are therefore advised to reduce the number of citations, perhaps 30 might be acceptable.

Thank you for this valuable comment. We agree that the number of references may seem to be high. Nevertheless, the studies, which are presented in literature, regarding coriander quality and influence of drying on it, is remarkably high. Our intention was to discuss obtained results comprehensively with earlier findings, to highlight that our study is a valuable contribution to the subject. Also, we want to avoid the claims that we pass over some earlier results, which may be interesting in light of our research. Moreover, the topic of drying is also broadly studied in numerous teams – the results between various  should to be confronted to continuously improve the overall picture of possibilities in medicinal and aromatic plants post-harvest treatment. We hope that our point of view will find the Reviewer understanding and acceptance.

Sincerely,

Jacek Łyczko, corresponding author.

Reviewer 3 Report

The manuscript entitled “Coriandrum sativum L. – Aroma Profile and Sensory Evaluation in Light of Multiple Drying Techniques” is a continuation of their previous studies regarding herbs and spices drying (Lavandula, O. Basilicum. From my point of view, the manuscript is more a determination of drying parameters of drying than scientific research. However, the manuscript is well organized and clearly written. I have several notes regarding statistical data processing.

  1. How was the homogeneity of the sample ensured? The study is not valid without this step.
  2. How many times was the drying experiment performed? In other words, what is the repeatability of the drying process? (at least the repeatability of the best conditions CPD70-VMFD and CD70? Repeatability is an essential parameter of each experiment and method.
  3. Page 5/line 7 – a procedure of the descriptive method is not clear. Why did every judge evaluate only 4 samples? Why not all 12 samples. How were the samples chosen?
  4. Page 5/the paragraph of “Statistical analysis”: The procedure of the napping test evaluation is missing.
  5. Table 3 – an explanation of the abbreviation LRI is missing
  6. Table 4 - How did the authors verify that the data for the Turkey test meet the requirements for the use of analysis of variance?

To conclude, a supplementing of this information is necessary for defending of experiment results.

Author Response

Jacek Łyczko, PhD candidate

Wrocław University of Environmental and Life Sciences

Department of Chemistry

Norwida 25, 50-375 Wrocław , Poland

jacek.lyczko@upwr.edu.pl

Dear Reviewer,

Thank you for your valuable comments regarding manuscript foods-1074621, entitled Coriandrum sativum L. – Aroma Profile and Sensory Evaluation in Light of Multiple Drying Techniques. We appreciate your detailed review and hope that our corrections will find your acceptance.

  1. How was the homogeneity of the sample ensured? The study is not valid without this step.

Thank you for this comment.  All purchased and used in the study plants were delivered on one day. Immediately, after the delivery, the coriander leaves were chopped down from all plants and thoroughly mixed together. The particular sample batches were taken from this created pool. This description is also given in Material and Methods section. Please see lines 97-99.

  1. How many times was the drying experiment performed? In other words, what is the repeatability of the drying process? (at least the repeatability of the best conditions CPD70-VMFD and CD70? Repeatability is an essential parameter of each experiment and method.

All drying variants were carried out in two technological repetitions. Information has been included in the Materials and Methods section. Please see lines 117-119.

  1. Page 5/line 7 – a procedure of the descriptive method is not clear. Why did every judge evaluate only 4 samples? Why not all 12 samples. How were the samples chosen?

During experiment design, we decided to reduce the numbers of samples subjected to descriptive sensory analysis, since 12 samples would be too much for panellists to perform a reliable evaluation. Of course in many cases, the panellists are confronted with 10 or more samples, nevertheless the cilantro sample presented very slight differences, therefore, to improve the sensory panel sensitivity, the number of samples was reduced.

The sample selection was based on ranking test results and napping results. For descriptive sensory analysis one sample from each kind of drying treatment/type, having high intensity in the ranking test  wand/or having no measurable intensity or low-intensity of off-flavours, were chosen.

Also, more information has been added to the materials and method section of the manuscript, to clarify the procedure of the descriptive sensory analysis. Please see changes in lines 201-207.

  1. Page 5/the paragraph of “Statistical analysis”: The procedure of the napping test evaluation is missing.

The statistical analysis of napping results was performed by XLSTAT Premium 2016 software. Detailed data analysis procedure was introduced into the Statistical analysis section. Please see lines 222-226.

  1. Table 3 – an explanation of the abbreviation LRI is missing

The explanation of LRI abbreviation is given with the very first use of it – please see line 173, therefore, we decided to not explain the abbreviation in table footer.

  1. Table 4 - How did the authors verify that the data for the Turkey test meet the requirements for the use of analysis of variance?

The data were introduced into the STATISTICA program and the heterogeneity of variance and the normality test were performed. Such approach was used in our earlier research, as well as other projects regarding the influence of drying on plant material and sensory analyses, carried out by particular co-authors of the manuscript.

Sincerely,

Jacek Łyczko, corresponding author.

Round 2

Reviewer 3 Report

No comment, OK

Author Response

Dear Reviewer,

thank you for acceptation of our improvments.

Kind regards,

Jacek Łyczko